# SAIL: Self-Amplified Iterative Learning for Diffusion Model Alignment with Minimal Human Feedback

**Xiaoxuan He**[1,2]*, **Siming Fu**[1]*†, **Wanli Li**[1], **Zhiyuan Li**[3], **Dacheng Yin**[2],
**Kang Rong**[2], **Fengyun Rao**[2], **Bo Zhang**[1]‡

[1] ZheJiang University,
[2] WeChat Vision, Tencent Inc
[3] Independent Researcher

## Abstract

Aligning diffusion models with human preferences remains challenging, particularly when reward models are unavailable or impractical to obtain, and collecting large-scale preference datasets is prohibitively expensive. *This raises a fundamental question: can we achieve effective alignment using only minimal human feedback, without auxiliary reward models, by unlocking the latent capabilities within diffusion models themselves?* In this paper, we propose **SAIL** (**S**elf-**A**mplified **I**terative **L**earning), a novel framework that enables diffusion models to act as their own teachers through iterative self-improvement. Starting from a minimal seed set of human-annotated preference pairs, SAIL operates in a closed-loop manner where the model progressively generates diverse samples, self-annotates preferences based on its evolving understanding, and refines itself using this self-augmented dataset. To ensure robust learning and prevent catastrophic forgetting, we introduce a ranked preference mixup strategy that carefully balances exploration with adherence to initial human priors. Extensive experiments demonstrate that SAIL consistently outperforms state-of-the-art methods across multiple benchmarks while using merely 6% of the preference data required by existing approaches, revealing that diffusion models possess remarkable self-improvement capabilities that, when properly harnessed, can effectively replace both large-scale human annotation and external reward models.

## 1 Introduction

Diffusion models have revolutionized generative AI, enabling the synthesis of high-fidelity images with remarkable diversity (Ramesh et al., 2022; Saharia et al., 2022; Rombach et al., 2022; Podell et al., 2023; Esser et al., 2024). However, aligning these models with human preferences remains a fundamental challenge, particularly in practical scenarios where reward models are unavailable or impractical to obtain (Black et al., 2023b; Fan et al., 2023; Clark et al., 2023; Wallace et al., 2024). This alignment problem becomes even more critical as diffusion models are increasingly deployed in real-world applications requiring nuanced understanding of human aesthetic and semantic preferences (Li et al., 2024; Hong et al., 2024).

Current approaches to preference alignment face a critical dilemma. Methods like DiffusionDPO (Wallace et al., 2024) achieve strong alignment but require massive human-annotated preference datasets—often millions of ranked pairs—making them prohibitively expensive and inflexible to evolving preferences. Alternatively, approaches utilizing external reward models (e.g., Aesthetic-based scorers (Black et al., 2023b; Fan et al., 2023)) introduce secondary biases and are vulnerable to reward hacking (Fu et al., 2025; Liu et al., 2024), while struggling with distributional shifts beyond their training data. ***Both paradigms create problematic dependencies***—either on exhaustive human

---

\* Equal Contribution.
† Project Leader.
‡ Corresponding authors.

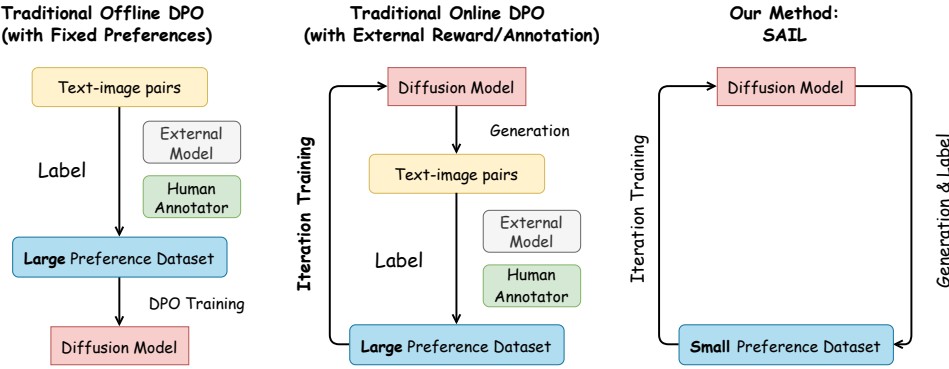

Figure 1: Comparsion of three direct preference optimization methods. Different from Offline DPO and Online DPO, SAIL iteratively update without large preference dataset and external reward model.

annotation efforts or on auxiliary models that may not generalize well—fundamentally limiting their practical applicability.

*This raises a crucial question: Can we achieve effective preference alignment using only minimal human feedback, without auxiliary reward models, by unlocking the latent alignment capabilities within diffusion models themselves?* We argue that diffusion models, once exposed to even a small set of human preferences, possess the inherent ability to act as their own teachers—progressively expanding their understanding through iterative self-improvement. This insight fundamentally reimagines the alignment process: rather than treating models as passive learners requiring constant external supervision, we can leverage their generative and discriminative capabilities in tandem. In this paper, we propose **SAIL** (**S**elf-**A**mplified **I**terative **L**earning), the ***first*** implicit self-rewarding framework that enables diffusion models to achieve strong preference alignment through autonomous bootstrapping. As illustrated in Figure 1, SAIL operates through a closed-loop learning process: starting from a minimal seed set of human-annotated preference pairs, the model iteratively generates diverse samples, self-annotates preferences based on its evolving understanding, and refines itself using this self-augmented dataset. The key innovation lies in our ***mathematical quantification*** of relative reward values between image pairs, enabling the diffusion model to serve dual roles as both generator and evaluator when conditioned on fixed reference parameters.

To ensure robust learning and prevent distribution collapse, we introduce a ranked preference mixup strategy that carefully balances exploration of the preference space with adherence to initial human guidance. This mechanism addresses the critical risk of catastrophic forgetting in self-training scenarios, maintaining alignment with human priors while enabling the model to discover nuanced preference patterns beyond the original annotations. Figure 2 demonstrates not only the effectiveness of our approach but also its remarkable stability over extended iterations. Our contributions to the community include:

- We propose SAIL, the first self-amplified iterative learning framework that enables diffusion models to achieve effective preference alignment through autonomous bootstrapping, eliminating dependencies on large-scale annotations and external reward models by developing a mathematical framework for self-reward quantification.

- We design a ranked preference mixup strategy that prevents catastrophic forgetting and ensures stable self-improvement, enabling the model to balance exploration of the preference space with adherence to human priors across extended iterations.

- Extensive experiments demonstrate that SAIL consistently outperforms state-of-the-art methods on HPSv2, Pick-a-Pic, and PartiPrompts benchmarks while using merely 6% of typical preference data, achieving superior qualitative results in texture and textual detail generation.

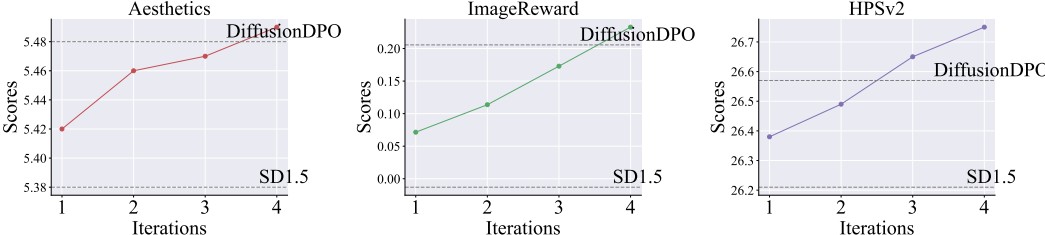

Figure 2: Iterative performance improvement with generated data of SAIL on Pick-a-Pic validation dataset in Aesthetics, ImageReward, and HPSv2. During the iterative process, SAIL demonstrates steady improvement and ultimately surpassed DiffusionDPO (as indicated by the dashed line).

## 2 RELATED WORK

### 2.1 HUMAN PREFERENCE OPTIMIZATION

Beyond mere visual fidelity, a critical frontier in text-to-image generation is aligning outputs with nuanced human preferences. A predominant strategy has been human-feedback–driven optimization, inspired by RLHF. For example, ImageReward + ReFL (Xu et al., 2023a) first train a reward model on 137K pairwise human comparisons and then fine-tune a diffusion model by backpropagating reward gradients, yielding substantial gains in aesthetic and caption alignment. Denoising Diffusion Policy Optimization (DDPO) (Black et al., 2023a) treat the denoising trajectory as an MDP and apply policy gradients to directly optimize black-box rewards such as CLIP similarity or aesthetic scores. DiffusionDPO (Wallace et al., 2024) align diffusion models to human preferences by directly optimizing on human comparison data. Building upon this framework, several advanced variants have emerged, MaPO (Hong et al., 2024) jointly maximizes the likelihood margin between preferred and dispreferred image sets, and the absolute likelihood of preferred samples. SPO (Liang et al., 2024) introduces employs a step-by-step optimization strategy, enabling finer control over localized quality improvements. The above methods rely on either large-scale preference datasets or pre-trained reward models. A critical yet understudied direction lies in self-alignment: leveraging the model's inherent generative capabilities to bootstrap preference learning without external supervision. **This capability is highly valuable in scenarios that require either a strong sense of realism or a distinct artistic style (no reward model)**.

### 2.2 ONLINE DIRECT PREFERENCE OPTIMIZATION

The Direct Preference Optimization (DPO) framework (Rafailov et al., 2023), initially for large language model alignment (Wu et al., 2025), directly refines policies with preference pairs without training an explicit reward network. A critical challenge in aligning diffusion models with human preferences lies in the inherent off-policy nature of conventional approaches: while the model continuously updates during training, the preference dataset is typically collected a priori, leading to a growing divergence between the model's current behavior and the static training data. Online AI feedback (Guo et al., 2024), uses an LLM as annotator: sample two responses from the current model and prompt the LLM annotator to choose which one is preferred, thus providing online feedback. Some methods eliminate the need for external annotators altogether by repurposing the DPO model itself as an implicit reward model (Kim et al., 2024; Chen et al., 2024; Cui et al., 2025), enabling iterative self-improvement. This approach is particularly appealing for diffusion models, where collecting high-quality preference data is inherently more challenging than in language tasks, and where a single reward model often fails to capture the nuanced, multi-dimensional aspects of image quality (e.g., composition, realism, and aesthetic appeal).

## 3 PRELIMINARY

**Direct preference optimization.** Direct Preference Optimization (DPO) (Rafailov et al., 2023) is a recently developed approach for aligning LLM $\pi_\theta$ with human preferences. The key idea behind DPO is to reparameterize the reward function in terms of the policy itself, eliminating the need for explicit

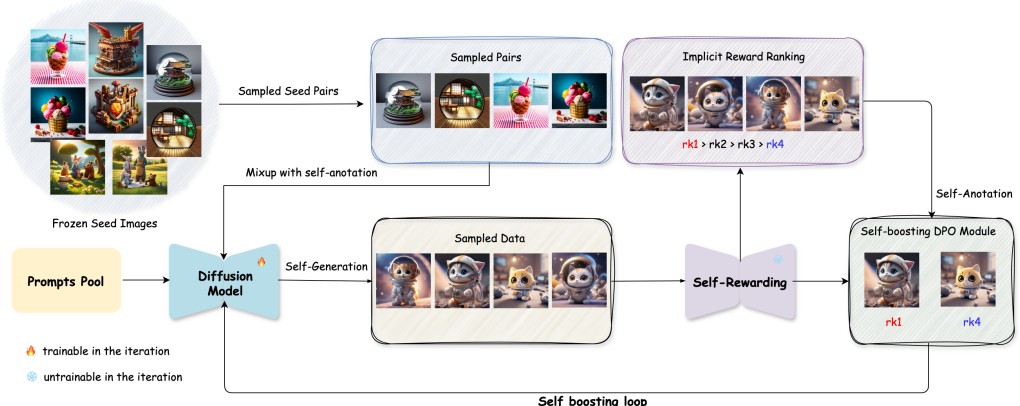

Figure 3: **Illustration of the proposed SAIL framework.** The SAIL framework incrementally refines the alignment of diffusion models through iterative cycles consisting of generating new preference data and conducting preference learning using mixup ranked preference data complemented by self-refinement mechanisms. This closed-loop self-boosting process operates with minimal initial data input, aiming to optimize performance by capitalizing on the intrinsic capabilities of the model, independent of external reward systems.

reward modeling. Specifically, the optimal reward function is derived from the RLHF objective, with the target LLM $\pi_\theta$ and the reference model $\pi_{\text{ref}}$.

$$r(\boldsymbol{y}, \boldsymbol{x}) = \beta \log \frac{\pi_\theta(\boldsymbol{x}|\boldsymbol{y})}{\pi_{\text{ref}}(\boldsymbol{x}|\boldsymbol{y})} + \beta \log Z(\boldsymbol{y}) \tag{1}$$

Then, the preference between two responses could be measured using this reward derivation, and $\pi_\theta$ is optimized to maximize this preference of $\boldsymbol{x}^w$ over $\boldsymbol{x}^l$ using the preference dataset $\mathcal{D}$.

$$p_\theta(\boldsymbol{x}^w > \boldsymbol{x}^l|\boldsymbol{y}) = \sigma(\beta \log \frac{\pi_\theta(\boldsymbol{x}^w|\boldsymbol{y})}{\pi_{\text{ref}}(\boldsymbol{x}^w|\boldsymbol{y})} - \beta \log \frac{\pi_\theta(\boldsymbol{x}^l|\boldsymbol{y})}{\pi_{\text{ref}}(\boldsymbol{x}^l|\boldsymbol{y})}) \tag{2}$$

$$L_{DPO}(\pi_\theta) = E_{(\boldsymbol{y}, \boldsymbol{x}^w, \boldsymbol{x}^l) \in \mathcal{D}}[-\log p_\theta(\boldsymbol{x}^w > \boldsymbol{x}^l|\boldsymbol{y})] \tag{3}$$

## 4 METHOD

***The key insight*** *is the development of a self-rewarding direct preference optimization framework, which is designed to iteratively maximize alignment with human preferences via closed-loop without any external reward model within few seed data.* Specifically, we start with a seed preference dataset $\mathcal{D}_{\text{init}} = \{(\boldsymbol{x}^w, \boldsymbol{x}^l, \boldsymbol{y})_n\}_{n=1}^N$ and a pre-trained diffusion model $\boldsymbol{\epsilon}_\theta^0$, i.e. SD1.5 or SDXL. The initial step involves fine-tuning $\boldsymbol{\epsilon}_\theta^0$ on $\mathcal{D}_{\text{init}}$ using the DiffusionDPO (Wallace et al., 2024) to update $\boldsymbol{\epsilon}_\theta^0$. Subsequent updates to the diffusion model are performed iteratively, leveraging self-generated data and self-rewarding mechanisms to continually improve the model's performance. The overview framework is demonstrate in Figure 3.

### 4.1 SELF-REWARDING PERFERENCE RANKING WITH SELF-GENERATED DATA

Given the candidate images $\boldsymbol{x}^A$ and $\boldsymbol{x}^B$, the reward difference between two images is shown as Equation 2. The Equation 2 can be derive as:

$$p_\theta(\boldsymbol{x}^A > \boldsymbol{x}^B|\boldsymbol{y}) = \sigma(r(\boldsymbol{y}, \boldsymbol{x}^A) - r(\boldsymbol{y}, \boldsymbol{x}^B)) \tag{4}$$

In diffusion models, the optimal reward function can be derived as:

$$r(\boldsymbol{y}, \boldsymbol{x}) = \beta \log \frac{p_\theta(\boldsymbol{x}_0|\boldsymbol{y}, t, q_t(\boldsymbol{x}_0))}{p_{\text{ref}}(\boldsymbol{x}_0|\boldsymbol{y}, t, q_t(\boldsymbol{x}_0))} + \beta \log Z(\boldsymbol{y}, t, q_t(\boldsymbol{x}_0)) \tag{5}$$

---

**Algorithm 1**

---

**Input:** Diffusion Model $\epsilon_\theta^0$, seed preference dataset $\mathcal{D}_{init}$, number of improving iterations $T$, new prompt sets $\{Y_i\}_{i=1}^T$,

---

Obatin Human preference model $\epsilon_\theta^0$ using Diffusion-DPO with $\epsilon_\theta^1$ and $\mathcal{D}_{init}$
**for** $i = 1$ **to** $T$ **do**
    **Candidate Generation**. Sample $N$ candidate images $(\boldsymbol{x}^{(1)}, ..., \boldsymbol{x}^{(N)})$ from the model $\epsilon_\theta^i$.
    **Self-Rewarding Ranking**. Rank the candidate images and choose the best image and the worst image in $N$ images to construct paired preference data $\mathcal{D}_i$ with $\epsilon_\theta^i$ and $\epsilon_\theta^0$.
    **Mixup Ranked Preference Data.** Mix the generated data with seed preference dataset. $\mathcal{D}_i = \alpha\mathcal{D}_i + (1-\alpha)\mathcal{D}_{init}$
    **Closed-loop Boosting Preference Optimization.** Update the current model with Eq. 3 and obtain $\epsilon_\theta^{i+1}$.
**end for**
**return** $\epsilon_\theta^{T+1}$

---

where $t$ is the timestep and $q_t(\boldsymbol{x}_0) = \sqrt{\alpha_t}\boldsymbol{x}_0 + \sqrt{1-\alpha_t}\epsilon$ is the combination of $\boldsymbol{x}_0$ and $\epsilon \sim \mathcal{N}(0, \boldsymbol{I})$, $\alpha_t$ is noise scheduler. With the noise prediction $\epsilon_\theta$ of diffusion models, we can derive and simplify the term $p_\theta(\boldsymbol{x}_0|\boldsymbol{y}, t, q_t(\boldsymbol{x}_0))$:

$$p_\theta(\boldsymbol{x}_0|\boldsymbol{y}, t, q_t(\boldsymbol{x}_0)) = \mathcal{N}(\boldsymbol{x}_0; \frac{q_t(\boldsymbol{x}_0) - \sqrt{1-\alpha_t}\epsilon_\theta}{\sqrt{\alpha_t}}, \delta_t\boldsymbol{I}) \tag{6}$$

$$\approx e^{-\frac{\delta_{t+1}^2}{2\delta_t^2}||\epsilon - \epsilon_\theta||^2} \tag{7}$$

Since $\log Z(\boldsymbol{y}, t, q_t(\boldsymbol{x}_0))$ is the same for a fixed prompt $\boldsymbol{y}$. The reward function for image $\boldsymbol{x}^A$ can be formulated:

$$r(\boldsymbol{y}, \boldsymbol{x}^A) \approx -\frac{\beta}{2}(||\epsilon^A - \epsilon_\theta(\boldsymbol{x}_t^A, \boldsymbol{y}, t)||_2^2 - ||\epsilon^A - \epsilon_{\text{ref}}(\boldsymbol{x}_t^A, \boldsymbol{y}, t)||_2^2) \tag{8}$$

Finally, we can apply the Equation 8 in Equation 4 and obtain the following term to judge the **relative** reward value of images $\boldsymbol{x}^A$ and $\boldsymbol{x}^B$:

$$p_\theta(\boldsymbol{x}^A > \boldsymbol{x}^B|\boldsymbol{y}) = \sigma(-\frac{\beta}{2}((||\epsilon^A - \epsilon_\theta(\boldsymbol{x}_t^A, \boldsymbol{y}, t)||_2^2 - ||\epsilon^A - \epsilon_{\text{ref}}(\boldsymbol{x}_t^A, \boldsymbol{y}, t)||_2^2)-$$
$$(||\epsilon^B - \epsilon_\theta(\boldsymbol{x}_t^B, \boldsymbol{y}, t)||_2^2 - ||\epsilon^B - \epsilon_{\text{ref}}(\boldsymbol{x}_t^B, \boldsymbol{y}, t)||_2^2))) \tag{9}$$

In enhancing the precision of estimates, it proves beneficial to average across multiple samples, denoted as $(t, q_t(\boldsymbol{x}_0))$. We compute estimates based on 10 random draws of $(t, q_t(\boldsymbol{x}_0))$. Consequently, the assignment of preference labels to the tuple $(\boldsymbol{x}^A, \boldsymbol{x}^B)$ is governed by the following formulation:

$$(\boldsymbol{x}^w, \boldsymbol{x}^l) = (\boldsymbol{x}^A, \boldsymbol{x}^B) \text{ if } p_\theta(\boldsymbol{x}^A > \boldsymbol{x}^B \mid \boldsymbol{y}) > 0.5 \text{ else } (\boldsymbol{x}^B, \boldsymbol{x}^A) \tag{10}$$

We choose the best image and the worst image in $N$ images to construct paired preference data. Following the construction of the dataset $\mathcal{D}_i$, the $i$-th iteration of preference learning is executed by fine-tuning the diffusion model $\epsilon_\theta^i$. This training on the self-generated dataset $\mathcal{D}_i$ is aimed at enhancing alignment by propagating the human preference priors encapsulated in $\mathcal{D}_0$ through the capabilities of the diffusion model. **More details are in Appendix A.**

### 4.2 CLOSED-LOOP BOOSTING DIFFUSION MODEL WITH MIXUP RANKED PREFERENCE DATA

Despite these advantages, there exists the risk of distributional collapse and overfitting to synthetic data during iterative self-improvement. To address these challenges, we propose an enhancement to the preference learning methodology by integrating an *mixup ranked preference data strategy* inspired by experience replay (Zhang & Sutton, 2017) in reinforcement learning, designed to stabilize the learning process against such perturbations and ensure robust preference alignment.

Especially, for the $i$-th iteration ($i = 1, \ldots$), we assume that the new prompt set $Y_i = \{y\}$ is available, i.e., $Y_i \cap Y_j = \emptyset$ for all $j = 0, \ldots, i-1$. As summarized in Algorithm 1. During each iteration $i$,

Table 1: Comparison with other methods on SD1.5 and SDXL. For HPSv2, we report the score in Anime, Concept-Art, Painting and Photo. For Pick-a-Pic v2, we apply PickScore, ImageReward, Aesthetics and HPSv2 metrics to evaluate all methods. The best results are highlighted in **red** and the performance gain are highlighted in **boldface**.

| Model | Method | Data | HPSv2 | | | | Pick-a-Pic V2 | | | |
| | | | Ani. | Con. | Paint. | Photo | P.S. | I.R. | Aes. | HPSv2 |
|---|---|---|---|---|---|---|---|---|---|---|
| SD1.5 | - | - | 27.23 | 26.65 | 26.52 | 27.41 | 20.62 | -0.0130 | 5.38 | 26.21 |
| | DiffusionDPO | 0.8M | 27.64 | 26.97 | 26.90 | 27.56 | 21.07 | 0.2056 | 5.48 | 26.57 |
| | DiffusionSPO | 0.8M | 28.05 | 27.49 | 27.61 | 27.55 | 21.20 | 0.1577 | 5.68 | 26.75 |
| | SAIL (Iter0) | 0.05M | 27.41 | 26.75 | 26.72 | 27.50 | 20.80 | 0.0715 | 5.42 | 26.38 |
| | SAIL (Iter1) | 0.05M | 27.56 | 26.88 | 26.83 | 27.59 | 20.89 | 0.1137 | 5.46 | 26.49 |
| | SAIL (Iter2) | 0.05M | 27.75 | 27.06 | 27.03 | 27.74 | 20.95 | 0.1729 | 5.47 | 26.65 |
| | SAIL (Iter3) | 0.05M | 27.88 | 27.16 | 27.15 | 27.79 | 21.00 | 0.2329 | 5.49 | 26.75 |
| | | | +0.65 | +0.51 | +0.63 | +0.38 | +0.38 | +0.2459 | +0.11 | +0.54 |
| SDXL | - | - | 28.03 | 27.17 | 27.22 | 27.50 | 22.13 | 0.6891 | 6.04 | 26.80 |
| | DiffusionDPO | 0.8M | 28.71 | 27.75 | 27.82 | 27.89 | 22.59 | 0.9336 | 6.02 | 27.27 |
| | MaPO | 0.8M | 28.39 | 27.60 | 27.58 | 27.74 | 22.24 | 0.8227 | 6.16 | 27.05 |
| | SAIL (Iter0) | 0.05M | 28.41 | 27.50 | 27.49 | 27.76 | 22.40 | 0.8705 | 6.04 | 27.08 |
| | SAIL (Iter1) | 0.05M | 28.59 | 27.62 | 27.59 | 27.90 | 22.53 | 0.9355 | 6.08 | 27.21 |
| | SAIL (Iter2) | 0.05M | 28.74 | 27.80 | 27.78 | 28.09 | 22.51 | 0.9844 | 6.16 | 27.32 |
| | | | +0.71 | +0.63 | +0.56 | +0.59 | +0.38 | +0.2953 | +0.12 | +0.52 |

For each prompt $y \in Y_i$, we sample $N$ candidate images $(\boldsymbol{x}^{(1)}, ..., \boldsymbol{x}^{(N)})$ by utilizing the intrinsic generation and reward modeling capabilities of the diffusion models $\boldsymbol{\epsilon}_\theta^i$, where $\boldsymbol{\epsilon}_\theta^i$ is the resulting model from the previous iteration. Then, using the reward captured with $\boldsymbol{\epsilon}_\theta^i$ and $\boldsymbol{\epsilon}_\theta^0$ (Eq. 5), we measure the *relative* preference between $\boldsymbol{x}^{(1)}$ and $\boldsymbol{x}^{(N)}$ and construct generated preference dataset $\mathcal{D}_i$. After that, we construct the mixed dataset $\mathcal{D}_i$ by sampling $\alpha$ proportion of data from $D_i$ and $(1 - \alpha)$ proportion of data from $\mathcal{D}_{init}$. Finally, DPO training is conducted on $D_i$ using $\boldsymbol{\epsilon}_\theta^i$ as both the initial policy and the reference policy, resulting in the updated model $\boldsymbol{\epsilon}_\theta^{i+1}$.

## 5 EXPERIMENTAL RESULTS

### 5.1 EXPERIMENT SETTINGS

**Implementation Details.** We demonstrate the effectiveness of SAIL across a range of experiments. We apply Stable Diffusion 1.5 (SD1.5) (Rombach et al., 2022) and Stable Diffusion XL-1.0 (SDXL) (Podell et al., 2023) as our base model. For preference learning dataset, we utilize Pick-a-Pic dataset (Kirstain et al., 2023), following the previous work. We use the larger Pick-a-Pic v2 dataset. After excluding the 12% of pairs with ties, we end up with 851,293 pairs, with 58,960 unique prompts. We first randomly select 50K preference data from the larger Pick-a-Pic v2 dataset, and then choose the remaining prompts from the remaining prompts for self improvement. We apply SAIL three iterations on SD1.5 and two iterations on SDXL, each iteration with 10K, 20K, 20K prompts. Morever, we set the mix ratio of human preference data and generated preference data as 0.25 in each iteration.

**Hyperparameters.** Following DiffusionDPO, We use AdamW for SD1.5 experiments, and Adafactor for SDXL to save memory. An effective batch size of 128 (pairs) is used. For image generation, we set $N = 8$ for quickly sampling. Morever, we apply DDPM with 50 steps for SD1.5 and DDIM with 20 steps in SDXL for quickly sampling in training and testing. All test images are generated classifier free guidance scale of 5 (SDXL) or 7.5 (SD1.5) during inference. For DPO training, we present the main SD1.5 and SDXL results with $\beta = 5000$.

**Evaluation.** We evaluate the proposed SAIL on three popular benchmarks: Pick-a-Pic, PartiPrompts and HPSv2. For Pick-a-Pic, We evaluate quantitative results based on the 500 validation prompts, i.e., validation unique. PartiPrompts contains 1,632 prompts encompassing various categories. Meanwhile,

HPSv2 comprises 3,200 prompts. covering four styles of image descriptions: animation, concept art, paintings and photo. For metrics, we use multiple evaluation metrics, indluding PickScore (general huamn preference) (Kirstain et al., 2023), Aesthetics (no-text-based visual appeal) (Meyer & Verrips, 2008), HPSv2 (prompt alignment) (Wu et al., 2023) and ImageReward (general human preference) (Xu et al., 2023b). *For all metrics, higher values indicate better performance.*

## 5.2 Primary Results: Aligning Diffusion Models

**Qualitative Comparison** Given the effectiveness and implementation efficiency of Direct Preference Optimization (DPO) (Rafailov et al., 2023), we adopt DPO as the foundational framework for iterative alignment. We compare SAIL with the base diffusion model (e.g., SD1.5, SDXL), vanilla DPO, and its variants (DiffusionSPO (Li et al., 2024), MaPO (Hong et al., 2024)) for fair comparison. DiffusionDPO and MaPO are training on Pick-a-Pic v2 dataset. Specically, SPO is different from the above method, which considering step-wise perference optimization not image-wise. Morever, SPO apply Pick-a-Pic to train a step-wise reward model. We demonstrate the quantitative result in Table 1. In HPSv2, experimental results demonstrate consistent performance improvement with increasing iterations, ultimately achieving a 0.71% (Anime), 0.63% (Concept Art), 0.56% (Painting), 0.59% (Photo) gain over the base model SDXL. Compared to DiffusionDPO with equivalent preference data, our method yields 0.33% (Anime), 0.30% (Concept Art), 0.29% (Painting), 0.33% (Photo) improvement in SDXL.

Remarkably, using only 6% human preference data (0.05M vs 0.8M samples), our approach surpasses the full-data DiffusionDPO baseline. Comprehensive evaluation on Pick-a-Pic dataset (measuring human preference, aesthetic quality, and text-image alignment) shows significant improvements across all four key metrics (0.38% in PickScore, 0.2953% in ImageReward, 0.12% Aesthetics, 0.52% HPSV2) in SDXL. Meanwhile, as illustrated in Table 1, our method robustly adapts to varying model scales (SD1.5 and SDXL), also achieving 0.38% in PickScore, 0.2459% in ImageReward, 0.11% Aesthetics, 0.54% HPSV2 in SD1.5, confirming its generalizability. In SD1.5, SAIL achieves similar performance with the DiffusionSPO, which uses a specific reward model and step-wise to align human preference. Table 1 further verifies that fully exploiting the model's intrinsic potential can achieve impressive human alignment without external data expansion.

Table 2: The performance of each iterations of SAIL in Partiprompts, Stable Diffusion 1.5. We report PickScore, Aesthetics, ImageReward and HPSv2 to evaluate the effectiveness of our method. The performance gain are highlighted in **boldface**.

| Model | P.S. | Aes. | I.R. | HPSv2 |
|---|---|---|---|---|
| SD1.5 | 21.37 | 5.26 | 0.1177 | 26.82 |
| SAIL (Iter0) | 21.48 | 5.28 | 0.1951 | 26.94 |
| SAIL (Iter2) | 21.54 | 5.31 | 0.2661 | 27.04 |
| SAIL (Iter3) | 21.62 | 5.34 | 0.3198 | 27.19 |
| SAIL (Iter4) | 21.61 | 5.36 | 0.3072 | 27.26 |
| | +0.24 | +0.10 | +0.1895 | +0.44 |

Meanwhile, we also evaluate SAIL in Partiprompts. Partiprompts can be used to measure model capabilities across various categories and challenge aspects. Partiprompts can be simple and can also be complex, which brings challenge in model evaluation. We conduct SAIL in SD1.5 and present the result in Table 2. Table 2 introduces the consistent performance gain in each score, 0.24% in PickScore, 0.1895% in ImageReward, 0.10% Aes, 0.44% HPSV2 improvement.

**Quantitative Comparison** As shown in Figure 4, SAIL demonstrates significant qualitative improvements over the base SDXL model. Quantitative results demonstrate generated data and self-rewarding can also achieve effective human preference alignment. SAIL achieves consistent visual improvement with the iteration increase. Quantitative experiments demonstrate our method's significant improvements across structural coherence and aesthetic quality.

## 5.3 Initial on Large Seed Data

We conducted experiments to explore initializing the model with more data. Specifically, we used the entire Pick-a-Pic v2 training dataset for initialization and selected prompts from JournyDB (Sun et al., 2023) as our subsequent prompt pool. Since our iter0 model is essentially DiffusionDPO, we directly continue training from this baseline. The experimental results are presented in the Table 3.

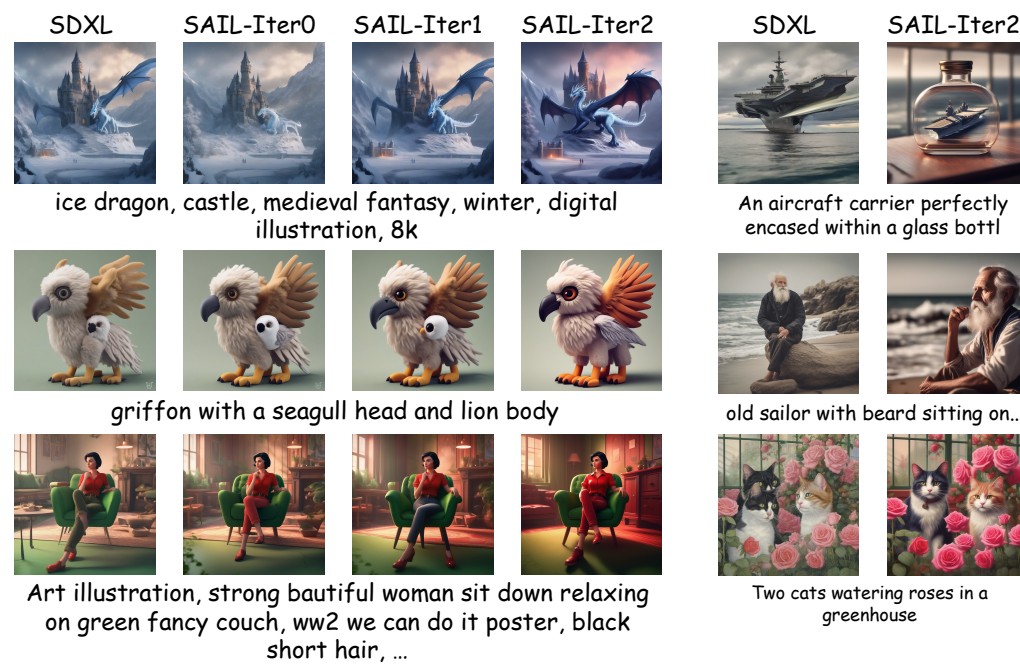

Figure 4: The qualitative results demonstrate the effectiveness of our method.

Table 3: Comparison with other methods on SD1.5. We build our SAIL on DiffusionDPO and mark as SAIL*. For HPSv2, we report the score in Anime, Concept-Art, Painting and Photo. For Pick-a-Pic v2, we apply PickScore, ImageReward, Aesthetics and HPSv2 metrics to evaluate all methods. The best results are highlighted in **red**.

| Model | Method | Data | HPSv2 | | | | Pick-a-Pic V2 | | | |
| | | | Ani. | Con. | Paint. | Photo | P.S. | I.R. | Aes. | HPSv2 |
| --- | --- | --- | --- | --- | --- | --- | --- | --- | --- | --- |
| SD1.5 | - | - | 27.23 | 26.65 | 26.52 | 27.41 | 20.62 | -0.0130 | 5.38 | 26.21 |
| | DiffusionDPO | 0.8M | 27.64 | 26.97 | 26.90 | 27.56 | 21.07 | 0.2056 | 5.48 | 26.57 |
| | DiffusionSPO | 0.8M | 28.05 | 27.49 | 27.61 | 27.55 | 21.20 | 0.1577 | 5.68 | 26.75 |
| | SAIL* (Iter1) | 0.8M | 27.86 | 27.15 | 27.08 | 27.65 | 21.16 | 0.2761 | 5.56 | 26.68 |
| | SAIL* (Iter2) | 0.8M | 28.08 | 27.37 | 27.31 | 27.72 | 21.27 | 0.4303 | 5.60 | 26.81 |

We apply SAIL* for two iterations and each iterations with 10K, 20K prompts. Table 3 demonstrates the result of SAIL* compared with SD1.5, DiffusionDPO and DiffusionSPO (Liang et al., 2024). On the Pick-a-Pic validation set, SAIL* outperforms DiffusionSPO across three metrics, most notably on ImageReward, where we achieve a performance of 0.4303%. Furthermore, on the HPSv2 dataset, our method also surpasses DiffusionSPO in two subclasses, Anime and Photo.

## 5.4 COMPARSION WITH ONLINE DPO

As illustrated in Figure 1, in LLM, existing approaches employ external models for online DPO optimization, but their core limitation lies in heavy reliance on extensive annotated data to train robust reward models. Taking DDPO (Black et al., 2023b) as an example, this method requires concurrently training four separate model (Aesthetic/Compressibility/Incompressibility/Prompt-Image Alignment) to comprehensively evaluate human preferences. Our experiments follow the setting in DDPO's framework (using only Aesthetic as the single reward model). Table 4 presents the comparsion between Online DPO and SAIL. While OnlineDPO-Aes achieves remarkable improvement in aesthetic metrics, which surpass SAIL by 0.07%. But its gains in human preference and text-image alignment remain limited. This confirms that single reward model struggle with comprehensive enhancement, including human preference, aesthetic quality, and text-image alignment — **though the ideal solution may be introducing multi reward models, how to balance their weight presents new research challenges**. In contrast, our method demonstrates dual advantages: Eliminates dependency on external annotations through self-rewarding based closed-loop optimization; Balanced Performance:

Table 4: Comparsion with Online DPO in Iter1. Base on the Iter0 model, we apply self-rewarding and Aesthetics rewarding to rank preference pairs given the generated data.

| Model | I.R. | Aes. | HPSv2 |
|---|---|---|---|
| SD1.5 | -0.0130 | 5.38 | 26.21 |
| Iter0 | 0.0715 | 5.42 | 26.38 |
| Self Rewarding & Aesthetics Rewarding | | | |
| SAIL | 0.1137 | 5.46 | 26.49 |
| OnlineDPO | 0.0936 | 5.53 | 26.35 |

Table 5: Comparsion of different selection strategy in constructing preference data of Iter1. Base on the Iter0 model, we apply Best-worst and random to choose win-lose pairs.

| Model | I.R. | Aes. | HPSv2 |
|---|---|---|---|
| SD1.5 | -0.0130 | 5.38 | 26.21 |
| Iter0 | 0.0715 | 5.42 | 26.38 |
| Select Strategy | | | |
| best-worst | 0.1137 | 5.46 | 26.49 |
| random | 0.1055 | 5.44 | 26.40 |

Achieves consistent improvements across all metrics. Morever, Pure aesthetic optimization may cause aesthetic overfitting (e.g., oversaturated colors), SAIL outputs better align with composite human preferences.

## 5.5 ABLATION STUDY

**Pair Selection Strategy.** We compare two pair-selection methods for constructing preference data from N candidates: (1) Best-worst Selection: select the best and the worst sample to construct preference data; (2) Randomized Selection: randomly select two samples and construct preference data. We conduct experiments and present the result in Table 5. Table 5 shows that SAIL achieves the best performance when using Best-worst Selection, 0.1137% in ImageReward, 5.46% in Aesthetic and 26.49% in HPSv2.

**The role of Mixup Ranked Preference Data.**
We reveal the critical role of mixing generated and human preference data in maintaining model stability. As shown in Table 6 , SAIL without mixed data suffers from a significant performance drop in the second iteration, attributed to two key factors: (1) *Overfitting to High-Confidence Pairs*: The model overfits to high-reward sample pairs during training phrase, drastically reducing generation diversity. For instance, images generated from different seeds become similar, while reward scores artificially inflate (e.g., exceeding 90%). (2) *Catastrophic*

Table 6: The influence of Mixup Ranked Preference Data.

| Model | P.S. | I.R. | HPSv2 |
|---|---|---|---|
| Base | 20.62 | -0.0130 | 26.21 |
| SAIL (Iter0) | 20.80 | 0.0715 | 26.38 |
| SAIL (Iter1) | 20.89 | 0.1137 | 26.49 |
| Iter2 | 20.95 | 0.1729 | 26.65 |
| Iter2 w/o mix | 20.86 | 0.1564 | 26.55 |

*Forgetting*: The absence of human preference data leads to progressive degradation of the model's discriminative ability. Compared to the first iteration, the reward model's accuracy declines measurably, which incurs the generated preference pairs is not accuracy. More discussion about the role of Mixup Ranked Preference Data is in Appendix.

## 6 CONCLUSION

In this work, we propose SAIL, a novel self-rewarding framework for aligning diffusion models with human preferences without relying on large-scale annotated datasets or external reward models. By leveraging iterative self-improvement through closed-loop generation and preference learning, SAIL effectively expands limited human seed annotations into robust alignment signals. Our approach addresses key limitations of existing methods—costly data dependency and bias propagation—while introducing mixup-ranked preference data to mitigate catastrophic forgetting and stabilize training. Experiments demonstrate that SAIL outperforms state-of-the-art methods even with only 6% of human preference data, highlighting its efficiency and scalability.

**Limitations.** We primarily focus on leveraging a small amount of human preference data and model-generated data for human preference alignment. However, compared to the image domain, preference data in video generation is significantly harder to collect. **We therefore foresee a promising future for exploiting SAIL in video human preference alignment and investigating intermediate reward mechanisms that can provide step-wise guidance throughout the denoising process.**

**Use of LLMs.** We utilize LLMs to assist with experimental design and writing refinement.

## ACKNOWLEDGEMENTS

This work was partially supported by the National Natural Science Foundation of China under Grant No. 62402434.

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

## A    MORE DETAILS ABOUT SELF-REWARDING STRATEGY

According to the Equation 5 in the main paper, which is derived in the DiffusionDPO Wallace et al. (2024). We have the reward function for images $x$ as follows:

$$r(\boldsymbol{y}, \boldsymbol{x}) = \beta \log \frac{p_\theta(\boldsymbol{x}_0|\boldsymbol{y}, t, q_t(\boldsymbol{x}_0))}{p_{\text{ref}}(\boldsymbol{x}_0|\boldsymbol{y}, t, q_t(\boldsymbol{x}_0))} + \beta \log Z(\boldsymbol{y}, t, q_t(\boldsymbol{x}_0)) \tag{11}$$

where $t$ is the timestep and $q_t(\boldsymbol{x}_0) = \sqrt{\alpha_t}\boldsymbol{x}_0 + \sqrt{1-\alpha_t}\boldsymbol{\epsilon}$ is the combination of $\boldsymbol{x}_0$ and $\boldsymbol{\epsilon} \sim \mathcal{N}(0, \boldsymbol{I})$, $\alpha_t$ is noise scheduler. Following DDIM Song et al. (2020), The $p_\theta(\boldsymbol{x}_0|\boldsymbol{y}, t, q_t(\boldsymbol{x}_0))$ can be dirve as:

$$\begin{aligned}
p_\theta(\boldsymbol{x}_0|\boldsymbol{y}, t, q_t(\boldsymbol{x}_0)) &= \mathcal{N}(\boldsymbol{x}_0; \boldsymbol{x}_0^{pred}, \delta_t \boldsymbol{I}) \\
&= \mathcal{N}(\boldsymbol{x}_0; \frac{q_t(\boldsymbol{x}_0) - \sqrt{1-\alpha_t}\boldsymbol{\epsilon}_\theta}{\sqrt{\alpha_t}}, \delta_t \boldsymbol{I}) \\
&= \mathcal{N}(\boldsymbol{x}_0; \boldsymbol{x}_0 + \frac{\sqrt{1-\alpha_t}\boldsymbol{\epsilon} - \sqrt{1-\alpha_t}\boldsymbol{\epsilon}_\theta}{\sqrt{\alpha_t}}, \delta_t \boldsymbol{I}) \\
&= \frac{1}{(2\pi\delta_t^2)^{(d/2)}} e^{-\frac{1}{2\delta_t^2}(\boldsymbol{x}_0 - (\boldsymbol{x}_0 + \frac{\sqrt{1-\alpha_t}}{\sqrt{\alpha_t}}(\boldsymbol{\epsilon}-\boldsymbol{\epsilon}_\theta)))^2} \\
&= \frac{1}{(2\pi\delta_t^2)^{(d/2)}} e^{-\frac{1}{2\delta_t^2}(\frac{\sqrt{1-\alpha_t}}{\sqrt{\alpha_t}})^2||\boldsymbol{\epsilon}-\boldsymbol{\epsilon}_\theta||^2} \\
&= \frac{1}{(2\pi\delta_t^2)^{(d/2)}} e^{-\frac{1-\alpha_t}{2\alpha_t\delta_t^2}||\boldsymbol{\epsilon}-\boldsymbol{\epsilon}_\theta||^2} \\
&= \frac{1}{(2\pi\delta_t^2)^{(d/2)}} e^{-\frac{\delta_{t+1}^2}{2\delta_t^2}||\boldsymbol{\epsilon}-\boldsymbol{\epsilon}_\theta||^2}
\end{aligned} \tag{12}$$

In Equation 12, $\frac{\delta_{t+1}^2}{\delta_t^2} \approx 1$. To evaluate the effectiveness of SAIL, we demonstrate the accuracy of several reward methods (PickScore Kirstain et al. (2023), Aesthetics Meyer & Verrips (2008), Clip Radford et al. (2021) and Self-rewarding). We choose 1,000 preference data from Pick-a-Pic v2 validation set and take the human preference as the ground truth. Figure 5 shows the results. $\text{SR}^1$, $\text{SR}^2$, $\text{SR}^3$ represent different versions of self-rewarding method in Iter0. The main difference between the above methods is the number of $(t, q_t)$, $\text{SR}^1$, $\text{SR}^2$, $\text{SR}^3$ base on 10, 20, 30 draws, respectively. It demonstrates the only with 0.05M data, the self-rewarding of iter0 surpasses the Clip and Aesthetics.

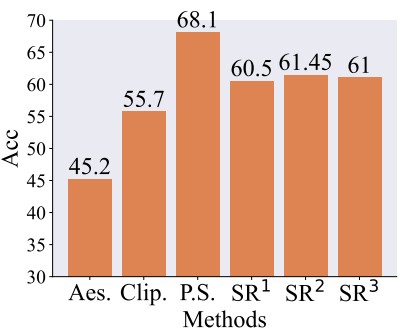

Figure 5: Comparsion of five reward models.

## B    VISUALIZATION OF SAIL*

we present visualizations comparing outputs from SD1.5, DiffusionDPO, and our model after SAIL* iterations. These visualizations demonstrate that after SAIL* iterations in Figure 6, the generated images exhibit higher quality, improved text-image alignment, enhanced aesthetics, and better overall texture.

## C    THE NUMBER OF CANDIDATES $N$

SD1.5  DiffusionDPO SAIL*-Iter1 SAIL*-Iter2

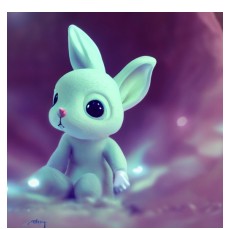 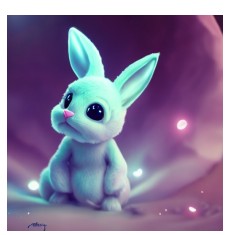 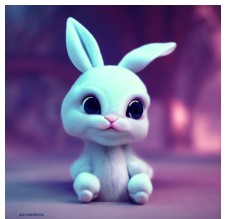 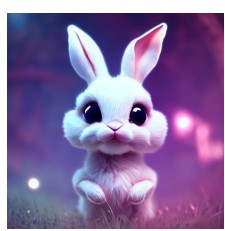

Cute and adorable cartoon rabbit baby rhea facing the camera,
fantasy, dreamlike, surrealism, super cute, trending on artstationm
volumetric light, cinematic, post processing, 8K

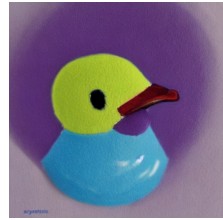 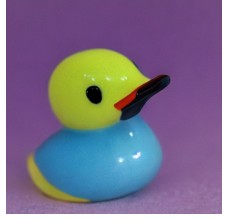 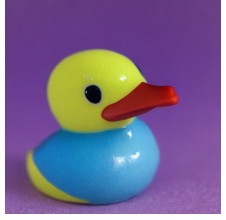 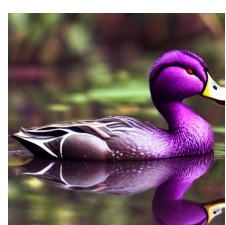

A small purple duck

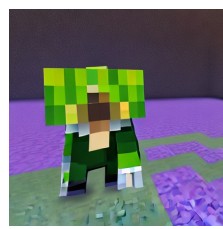 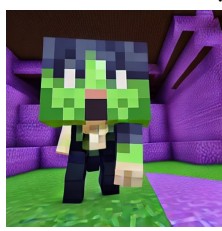 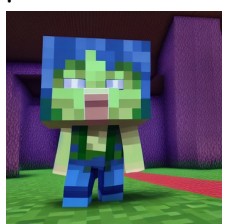 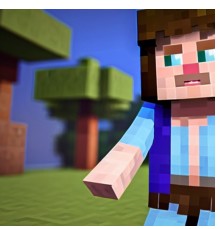

A Minecraft character named Herobrine standing on a grass block

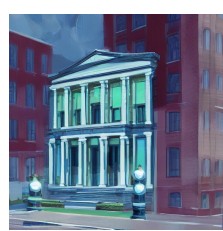 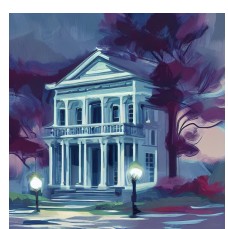 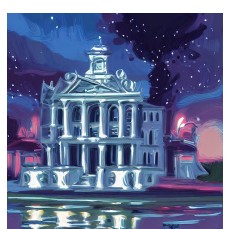 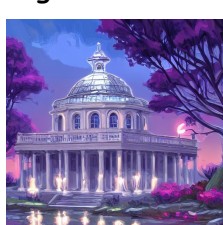

A landscape featuring a unique digital painting-style building

Figure 6: Qualitative visual results for several methods.

Table 7: The influence of Candidates $N$.

| Model | P.S. | I.R. | HPSv2 |
|---|---|---|---|
| Base | 20.62 | -0.0130 | 26.21 |
| Candidates $N$ | | | |
| 2 | 20.70 | 0.0536 | 26.35 |
| 4 | 20.78 | 0.0879 | 26.40 |
| 8 | 20.89 | 0.1137 | 26.49 |
| 16 | 20.96 | 0.1356 | 26.54 |

We conducted experiments to determine the optimal number of candidate images per iteration. The results, shown in Table 7, illustrate the impact of varying candidate image counts on the outcome of the first iteration. It is evident that increasing the number of candidate images significantly improves the results. However, considering both image generation efficiency and performance, we opted to use $N = 8$ in our study.

Table 8: The performance of SAIL with different seed preference dataset on SD1.5.

| Method | Pick-a-Pic V2 | | | |
| --- | --- | --- | --- | --- |
| | P.S. | I.R. | Aes. | HPSv2 |
| - | 20.62 | -0.0130 | 5.38 | 26.21 |
| SAIL (seed1) | 21.00 | 0.2329 | 5.49 | 26.75 |
| SAIL (seed2) | 21.05 | 0.2381 | 5.50 | 26.79 |
| SAIL (seed3) | 21.02 | 0.2253 | 5.49 | 26.76 |

## D    THEORETICAL ANALYSIS

Theoretically, SAIL shares the same global objective as standard DPO: finding the policy $\pi^*$ that maximizes the reward (the underlying human preference) subject to a KL-divergence constraint. SAIL (Iterative/Semi-Supervised) can be viewed as a Self-Training process. The model explores the latent space to generate new samples and uses its current implicit reward function to estimate rankings (pseudo-labels). This propagates the reward signal into previously unexplored regions.

Therefore, SAIL converges to a distribution that is consistent with human preferences over a much broader support set than standard DPO. It approximates the optimal solution of DPO over the full data distribution, rather than just the limited set.

## E    SAIL ON DIFFERENT INITIAL DATASET

Table 8 indicates that SAIL is reasonably robust to seed selection. The Ranked Preference Mixup strategy plays a crucial role here. By mixing generated preferences with the high-quality seed data, SAIL prevents the model from drifting too far if the self-generated signals are noisy in the early stages.

## F    SAIL WITH MORE ITERATIONS

We perform SAIL with more iterations on SD1.5, as shown in Table 9. In iter 1-3, there are rapid improvements. The model effectively learns to align with the seed preferences and generalizes to the unlabelled set. While in iter 4-5, we observed that the probability in Equation 4 exceeds 0.8, indicating that the model has approached a saturation state.

Table 9: The performance of SAIL with more iterations on SD1.5.

| Method | Pick-a-Pic V2 | | | |
| --- | --- | --- | --- | --- |
| | P.S. | I.R. | Aes. | HPSv2 |
| - | 20.62 | -0.0130 | 5.38 | 26.21 |
| SAIL (Iter0) | 20.80 | 0.0715 | 5.42 | 26.38 |
| SAIL (Iter1) | 20.89 | 0.1137 | 5.46 | 26.49 |
| SAIL (Iter2) | 20.95 | 0.1729 | 5.47 | 26.65 |
| SAIL (Iter3) | 21.00 | 0.2329 | 5.49 | 26.75 |
| SAIL (Iter4) | 21.04 | 0.2803 | 5.51 | 26.77 |
| SAIL (Iter5) | 21.06 | 0.2971 | 5.51 | 26.78 |

## G    VISUALIZATION

We present visual comparisons showcasing the effectiveness of SAIL across different base models.

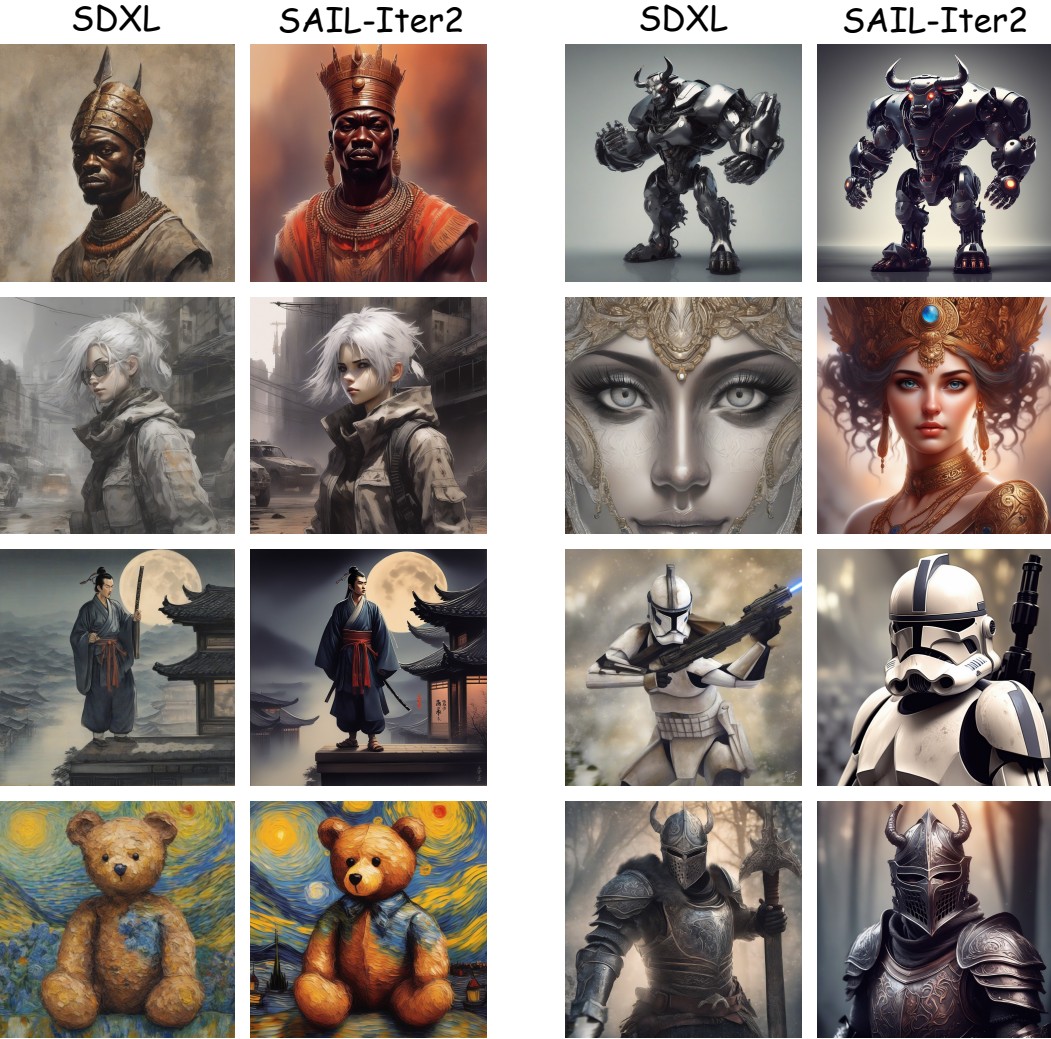

Figure 7: Qualitative visual analysis results for several methods. The images are presented row by row (top to bottom), and within each row, from left to right. The respective prompts are: (1) "A rich, ruthless and courageous warrior king from Africa", (2) "A robotic bull, full body shot, standing, athletic, steel and hard rubber, glowing eyes, humanoid, looking at camera, arms crossed", (3) "A girl with silver hair in a post apocalyptic setting portrayed in a cinematic illustration by Yoji Shinkawa and Krenz Cushart", (4) "beautiful goddess, detailed face, focus on eyes, masterpiece, realistic", (5) "A painting depicting a wuxia character standing on a roof under a moonlit night", (6) "Clone Trooper", (7) "A teddy bear inspired by Vincent van Gogh", (8) "The image is a stunning illustration of a knight warrior wearing Nordic armor and a Skyrim mask, with intricate details and dynamic lighting that make it perfect for RPG portraits and cosplay".

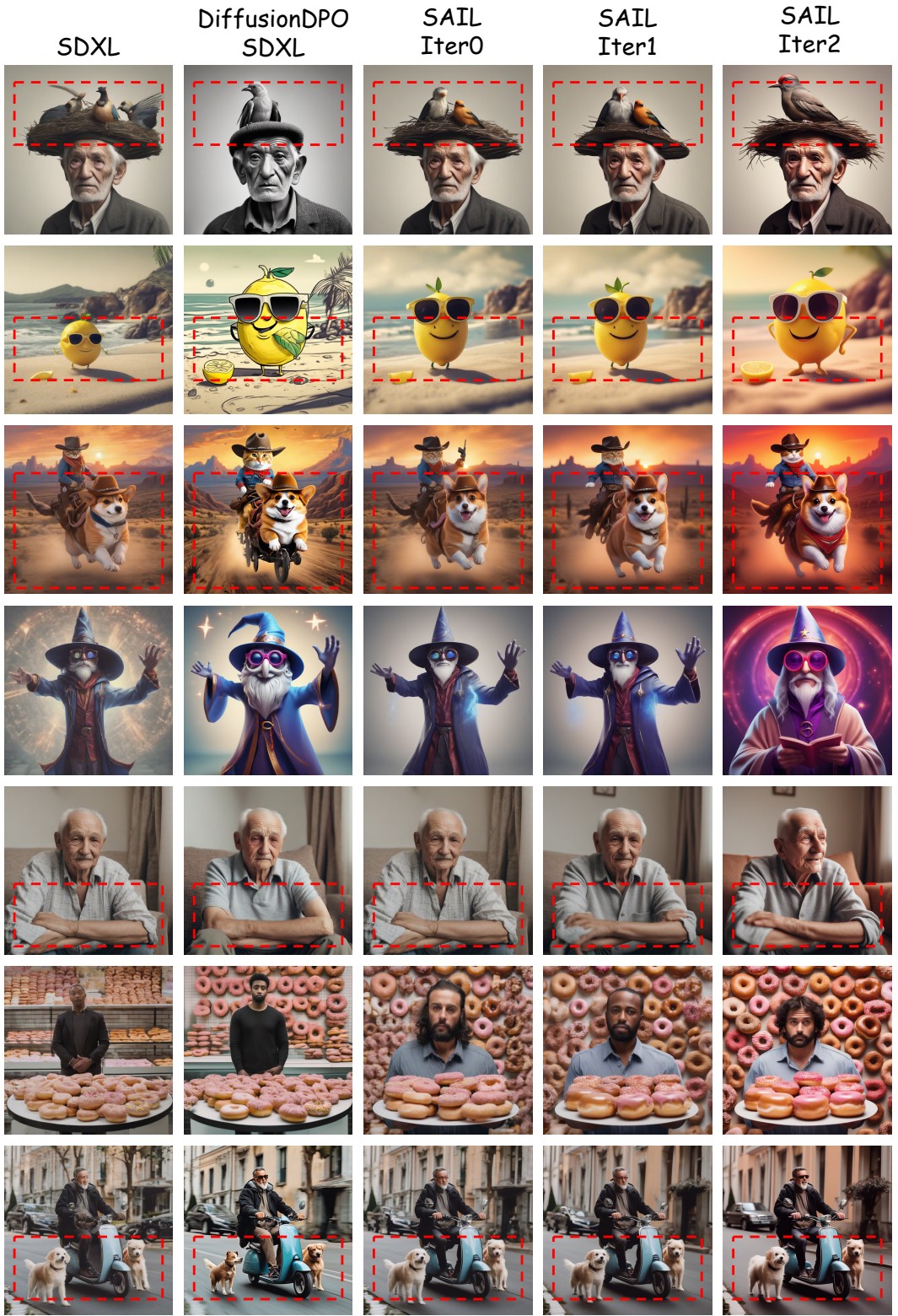

Figure 8: Qualitative visual analysis results for different methods. The prompts used for each row (from top to bottom) are as follows: (1) "An old man with a bird on his head", (2) "A lemon character with sunglasses on the beach", (3) "Cat wearing cowboy hat rides on corgi during sunset in the Wild West", (4) "There is an anthropomorphic male wizard in the image wearing 3D cinema glasses", (5) "An elderly man is sitting on a couch", (6) "A man standing in front of a bunch of doughnuts", (7) "A man and two dogs are riding a scooter".

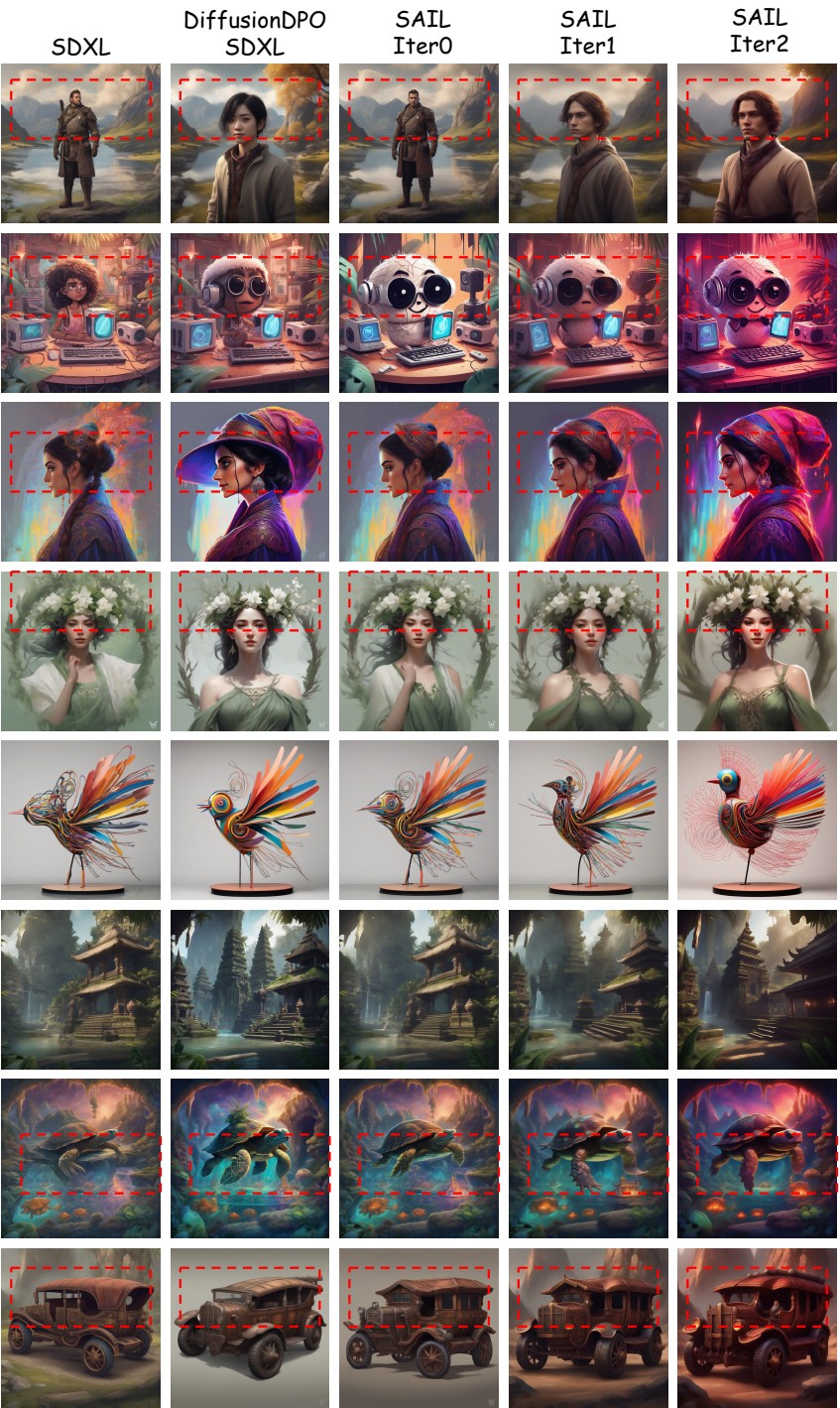

Figure 9: Qualitative visual analysis results for different methods. The prompts used for each row (from top to bottom) are as follows: (1) "A portrait of a character in a scenic environment", (2) "'A stylized portrait featuring sliced coconut, electronics, and AI in a cartoonish cute setting with a dramatic atmosphere", (3) "A side profile portrait of Maya Ali as a mage with intricate details, neon and sweat drops in a highly detailed digital painting", (4) "The image depicts a beautiful goddess of spring wearing a wreath and flowy green skirt, created by artist wlop", (5) "A kinetic sculpture of a colorful bird with a long tail surrounded by swirling lines and shapes", (6) "A concept art digital CG painting of a place in Bali, trending on ArtStation and created using Unreal Engine", (7) "The image portrays a surreal scene of a hybrid creature consisting of a great leviathan, cybernetic turtle, and cephalopod terrapin in a magical universe surrounded by a cozy hot springs, cave, forest, and lush plants amidst a luminous stellar sky", (8) "A realistic digital art depicting a dwarven automobile".

