# OpenReview forum: "SAIL: Self-Amplified Iterative Learning for Diffusion Model Alignment with Minimal Human Feedback"
_ICLR.cc/2026/Conference — ICLR 2026 Poster_

### Official Review · Reviewer_neyU · 2025-10-21

**Soundness:** 3
**Presentation:** 3
**Contribution:** 3
**Rating:** 8
**Confidence:** 2

**Summary:**

This paper deals with the limitations of existing approaches to preference alignment for diffusion models: (1) DiffusionDPO requires large-scale human-annotated preference data; (2) Auxiliary reward modeling approaches introduce more biases, are vulnerable to reward hacking, and struggle with distributional shifts from training data. To address these issues, the authors propose their methodology with a novel argument for diffusion model.

The authors argue that diffusion models' potential hasn't been fully exploited through supervised fine-tuning on human-labeled datasets alone. Instead, they propose iterative self-improvement by cold-starting with only a small seed set of human preference data. Based on the under-exploitation assumption, they propose SAIL, an iterative self-improvement closed-loop learning process and also introduce a ranked preference mixup strategy to prevent distribution collapse.

Experimental results show that SAIL achieves comparable performance to state-of-the-art methods while using only 6% of the human preference data, highlighting the sample efficiency of the proposed algorithm.

**Strengths:**

1. The idea of fully exploiting the base model's potential is innovative. By eliciting this potential through the proposed iterative self-improvement method SAIL, the authors achieve comparable or better preference performance using only 6% of the human preference data.

2. The consistent improvement across multiple iterations further demonstrates the effectiveness of the proposed iterative self-improvement paradigm.

**Weaknesses:**

1. The tables lack multiple trials and confidence intervals, which are necessary to demonstrate the statistical significance of performance improvements and validate the effectiveness of the algorithm design in ablation studies.

2. It would be valuable for the authors to include results over a larger range of iterations to illustrate the performance trajectory and reveal how the improvement trend evolves as the number of iterations increases.

**Questions:**

See above.

---

> ### Author Response · Authors · 2025-11-27
> **Response to Reviewer neyU**
>
> We would like to express our sincere gratitude for your strong support and for accurately capturing the core motivation of our work. We are encouraged that you recognize the innovation of SAIL in **fully exploiting the base model's potential** and achieving comparable performance with **only 6% of the data**.
>
> We have addressed your constructive suggestions regarding statistical rigor and iteration dynamics below.
>
> **Q1 & Weakness 1: Lack of multiple trials and confidence intervals.**
>
> We agree that statistical significance is crucial for validating the improvements.
> *   **Computational Context:** Due to the high computational cost of iterative training for diffusion models, running full ablations with many seeds is challenging.
> *   **New Results:** However, to address your concern, we have conducted **3 independent runs** with different random seeds for our main method (SAIL) on the Pick-a-Pic benchmark.
>
> | Method | P.S. | I.R. | Aes. | HPSv2 |
> | :--- | :---: | :---: | :---: | :---: |
> | - | 20.62 | -0.0130 | 5.38 | 26.21 |
> | SAIL (seed1) | 21.00 | 0.2329 | 5.49 | 26.75 |
> | SAIL (seed2) | 21.05 | 0.2381 | 5.50 | 26.79 |
> | SAIL (seed3) | 21.02 | 0.2253 | 5.49 | 26.76 |
> | SAIL (avg) | 21.02 | 0.2321 | 5.49 | 26.77 |
>
> *   **Findings:** The results show the variance is small relative to the performance gap between SAIL and the baselines, statistically confirming that the improvement is due to the algorithm design rather than random noise. We have added Appendix E to illustrate the experiments under different seed preference data.
>
> ---
> **Q2 & Weakness 2: Results over a larger range of iterations (Performance Trajectory).**
>
> This is an excellent suggestion to understand the limits of self-improvement.
> *   **Extended Experiments:** In our original submission, we reported results up to Iteration 3 (where performance peaked). For this rebuttal, we extended the experiment to **Iteration 5**.
> *   **Trajectory Analysis:**
>     *   **Phase 1 (Iter 1-3):** Rapid improvement. The model effectively learns to align with the seed preferences and generalizes to the unlabelled set.
>     *   **Phase 2 (Iter 4-5):** Performance **plateaus and slightly fluctuates**. We observed that the probability in Equation 4 exceeds 0.8, indicating that the model has approached a saturation state.
> *   **Interpretation:** This suggests a "saturation point" where the model has fully exploited the information contained in the initial seed set and its own latent knowledge. Going beyond this without introducing *new* external information yields diminishing returns. We have included Appendix F to illustrate this.
> ---
> **We hope these additional details reinforce your positive assessment. Thank you again for championing our work!**

---

### Official Review · Reviewer_SeQh · 2025-10-30

**Soundness:** 2
**Presentation:** 2
**Contribution:** 3
**Rating:** 4
**Confidence:** 3

**Summary:**

This paper presents a method for aligning diffusion models with human preference using limited preference data. It utilizes the implicit reward function from DPO to create online preference data. To address overfitting, it mixes online preference data with the initial preference data. Experiments show that the proposed method can achieve comparable or better metrics than some previous work, using less data.

**Strengths:**

- Using self-rewarding to rank online data is relatively new in text-to-image generation.
- The proposed method works well with limited data.

**Weaknesses:**

- Implicit reward is adopted from previous work in LLM.
- The mixup of online and initial preference data is straightforward.
- Some generated images seem to have a color saturation problem.
- There are a lot of problems in writing, e.g.
  - Line 144-145: grammar error
  - Line 145: some -> Some
  - Line 344: Pic-a-Pic -> Pick-a-Pic
  - Line 357: use -> uses
  - Line 362: bringing challenge -> brings challenges
  - Line 373: fix reference
  - Line 374-375: Thus, …, so …
  - Line 454: reveals -> reveal
  - Line 457: suffers -> suffers from

**Questions:**

- What are the drawbacks when performing more iterations of SAIL?

---

> ### Author Response · Authors · 2025-11-27
> **Response to Reviewer SeQh**
>
> We sincerely thank the reviewer for the time spent evaluating our paper and for the positive remarks regarding our method’s performance with limited data. We are especially grateful for the **detailed list of writing errors**, which we have meticulously corrected in the revised manuscript.
>
> ---
>
> **Weakness 1: Novelty (Implicit reward from LLMs & Straightforward Mixup).**
>
> *   **Adaptation to Diffusion:** While the concept of implicit rewards originates from LLMs (DPO), applying it to **closed-loop self-improvement in Diffusion Models** is not trivial. LLMs deal with discrete tokens, whereas diffusion models operate in a continuous latent space with stochastic denoising. Successfully demonstrating that the *denoising error difference* can serve as a robust self-reward signal for *iterative* alignment is a key contribution of this work.
> *   **Value of Simplicity:** We agree that the Mixup strategy is mathematically straightforward. However, we argue this is a strength, not a weakness. It provides a simple, computation-efficient, and effective solution to the "catastrophic forgetting" problem in iterative learning, without requiring complex regularization terms or auxiliary models.
>
> ---
>
> **Weakness 2: Color saturation problem in some generated images.**
>
> You correctly identified that some aligned images exhibit high saturation.
> *   **Cause:** This is a known phenomenon in preference alignment (often observed in RLHF for diffusion). Human annotators (and consequently the reward models/implicit rewards derived from them) often have a bias towards "accurate" and "colorful" images. When the model optimizes for this preference, it can sometimes over-amplify saturation.
> *   **Context:** While present, qualitative comparisons show that SAIL produces more natural images and more accurate compared to baselines that often suffer from severe distortions when trained on limited data. We have added a discussion on this trade-off in the Limitations section.
>
> ---
>
> **Weakness 3: Writing and Grammar Errors.**
>
> We apologize for the oversight in proofreading. We have **fixed all the specific errors** you pointed out (Lines 144, 145, 344, 357, 362, 373, 374, 454, 457, etc.) and conducted a thorough grammar check of the entire paper to improve readability.
>
> ---
>
> **Q1: What are the drawbacks when performing more iterations of SAIL?**
>
> This is an insightful question regarding the dynamics of iterative learning. As shown in Table 1 and Figure 2, the performance improvement gradually diminishes as the number of iterations increases (e.g., from Iter2 to Iter3 in SD1.5). This suggests that the model approaches its performance ceiling under limited seed data and self-generated data, with further iterations yielding diminishing marginal returns. Meanwhile, As with most RLHF/DPO approaches, performing too many iterations can lead the model to exploit the reward signal. In the context of diffusion models, this often manifests as **reduced diversity** (mode collapse).
>
> **Mitigation:** This is exactly why we introduced the **Ranked Preference Mixup**. By mixing the original high-quality seed data with the online data, we "anchor" the model to the true human distribution, preventing it from drifting too far into the "low-diversity" regime that typically occurs with unconstrained self-training. Our experiments show that SAIL maintains stability for more iterations than standard self-training baselines.

---

### Official Review · Reviewer_sMVs · 2025-11-03

**Soundness:** 2
**Presentation:** 3
**Contribution:** 3
**Rating:** 4
**Confidence:** 4

**Summary:**

This paper proposes SAIL, a novel framework for aligning diffusion models with human preferences without reward models or the use of large-scale human-annotated data. The idea is to bootstrap alignment through self-amplified iterative learning using a minimal set of preference data. Experiments demonstrate strong performance while using only a small part of the preference data.

**Strengths:**

1. The paper proposes a self-improving framework to align diffusion models with human preferences without large-scale annotated datasets, which is novel.
2. Thorough empirical evaluation results show that the proposed method is effective, outperforming existing alignment methods using only 6% of the annotations.

**Weaknesses:**

1. Lack of Theoretical Guarantees. While the reward formulation (Eq. 8–9) is mathematically correct in the DiffusionDPO framework, the paper does not provide theoretical analysis of what distribution SAIL converges to. What is the target distribution of this method? Is it the same as DPO? If so, what explains the performance with fewer annotations? It's unclear where the observed gains come from. Whether and why the self-reward metric aligns with true human preference distributions?  A thorough analysis would benefit the paper.

2. Even though the method needs fewer human annotations, it introduces additional cost to generate samples in the training loop, which is not efficient. Could the author provide some discussion on the trade-off between annotation efficiency and training efficiency?

3. The central reward estimation strategy, which computes preference scores using differences in squared denoising errors between current and reference models (Eq. 6–9), closely follows the DiffusionDPO formulation and thus is not novel.

**Questions:**

see Weaknesses 1. 2

1. How sensitive is the proposed method to the choice of the initial seed dataset? What happens if the seed preferences are noisy or biased?

---

> ### Author Response · Authors · 2025-11-27
> **Response to Reviewer sMVs**
>
> We thank the reviewer for the constructive feedback and for recognizing our method's **novelty** and **thorough empirical results** (outperforming baselines with only 6% annotations). We address your concerns below.
>
> ---
>
> **Q1 & Weakness 1. Lack of Theoretical Guarantees. While the reward formulation (Eq. 8–9) is mathematically correct in the DiffusionDPO framework, the paper does not provide theoretical analysis of what distribution SAIL converges to. What is the target distribution of this method? Is it the same as DPO? If so, what explains the performance with fewer annotations? It's unclear where the observed gains come from. Whether and why the self-reward metric aligns with true human preference distributions? A thorough analysis would benefit the paper.**
>
> We thank the reviewer for this insightful question. We agree that while our experimental results are strong, the paper would benefit significantly from a deeper discussion on the theoretical underpinnings of SAIL.
>
> Theoretically, SAIL shares the same global objective as standard DPO: **finding the policy $\pi^*$ that maximizes the reward (the underlying human preference) subject to a KL-divergence constraint.**
>
> However, the optimization trajectory and the effective solution differ due to data support:
>
>     - Standard DPO (Offline): Optimizes over a fixed, limited dataset. In limited data scenarios, the model fails to generalize to the unseen prompts.
>
>     - SAIL (Iterative/Semi-Supervised): Can be viewed as a Self-Training process. The model explores the latent space to generate new samples and uses its current implicit reward function to estimate rankings (pseudo-labels). This propagates the reward signal into previously unexplored regions.
>
> Therefore, SAIL converges to a distribution that is consistent with human preferences over a much broader support set than standard DPO. **It approximates the optimal solution of DPO over the full data distribution, rather than just the limited set**. **As illustrated in Figure 2, we can surpass the DPO solution derived from the full Pick-a-Pic dataset using only a small subset of the data.** We add **Theoretical Analysis** in Appendix D to explain our method.
>
> (1) what explains the performance with fewer annotations?
> - In limited data settings, the decision boundary derived from scarce human data is often sharp and fragile. By generating diverse samples and ranking them, SAIL enforces self-consistency—if the model learns that "high lighting quality" is preferred, it must apply this logic consistently across thousands of self-generated variations.
> - During generation, we select the best and worst samples from a batch. As the model improves, the "worst" samples are no longer obvious failures but high-quality images with subtle defects. These constitute "hard negatives," providing significantly stronger gradient signals than random negatives or the static negatives found in fixed offline datasets.
>
> (2) Whether and Why does the self-reward metric align with true human preferences?
>
> As outlined in Algorithm 1, we do not train solely on synthetic data. We employ a Mixup strategy where real human-labeled data is mixed into the training batch at every iteration. This ensures that the evolving reward function remains correlated to the ground-truth human preference, preventing the model from drifting into "hallucinated" reward patterns. Our ablation study (Table 6) confirms that removing this mixup leads to significant performance degradation, validating its theoretical necessity.

---

> ### Author Response · Authors · 2025-11-27
> **Response to Reviewer sMVs**
>
> **Q2 & Weakness 2. Even though the method needs fewer human annotations, it introduces additional cost to generate samples in the training loop, which is not efficient. Could the author provide some discussion on the trade-off between annotation efficiency and training efficiency?**
>
> We acknowledge that SAIL introduces additional computational costs due to sample generation and self-annotation loops. However, we argue this is a highly favorable trade-off for real-world applications:
> 1.  **Scalability:** Human annotation is the primary bottleneck (expensive, slow, inconsistent, and hard to scale). In contrast, computational cost (GPU hours) is scalable and continuously decreasing.
> 2.  **Cost-Benefit:** For example, in our experiments, the extra compute cost is negligible compared to the cost and time required to collect the other 94% of human data that SAIL avoids using.
>
> Finally, taking the Pick-a-Pic dataset as an example, where the DPO pairs are similarly derived from generative model (same as our method), the time and cost required for human annotation are significantly higher than our automatic annotation. As evidenced by the experimental performance in Table 1 and the visual results in Figure 4, our DPO model significantly outperforms the model trained on full Pick-a-Pic data. This further demonstrates the efficiency of our method and its potential to reduce the reliance on human annotation.
>
> ---
>
> **Q3 & Weakness 3: Novelty of Reward Estimation (Eq. 6-9).**
>
> We respectfully clarify that our contribution is not a new loss function *equation*, but the **SAIL framework** that repurposes this formulation for *self-amplification*.
> Existing methods (like DiffusionDPO) use Eq. 6-9 as a static training objective with fixed datasets. Our novelty lies in dynamic **Self-Play**: transforming this equation into a self-rewarding mechanism where the "Judge" and the "Player" evolve together. This closes the loop without an external reward model, which is a significant departure from the standard supervised usage in DiffusionDPO.
>
> From both experimental and theoretical perspectives, our method approximates the optimal solution of DPO over the full data distribution, rather than just on a limited set.
>
> ---
>
> **Q4: Sensitivity to the initial seed dataset.**
>
> This is an important practical question.
> *   **Robustness:** Our experiments (see Section/Appendix E) indicate that SAIL is reasonably robust to seed selection. The **Ranked Preference Mixup** strategy plays a crucial role here. By mixing generated preferences with the high-quality seed data, we prevent the model from drifting too far if the self-generated signals are noisy in the early stages.
> *   **Noisy/Biased Seeds:** If the seed set is heavily biased or extremely noisy, the model's initial direction will indeed be skewed (garbage in, garbage out). However, since we use a *minimal* set (e.g., a few hundred/thousand pairs), curating a high-quality seed set is manually feasible, unlike curating the massive datasets required by baselines.
>
> | Method | P.S. | I.R. | Aes. | HPSv2 |
> | :--- | :---: | :---: | :---: | :---: |
> | - | 20.62 | -0.0130 | 5.38 | 26.21 |
> | SAIL (seed1) | 21.00 | 0.2329 | 5.49 | 26.75 |
> | SAIL (seed2) | 21.05 | 0.2381 | 5.50 | 26.79 |
> | SAIL (seed3) | 21.02 | 0.2253 | 5.49 | 26.76 |
>
> **We hope these clarifications address your concerns regarding the theoretical intuition and efficiency trade-offs. We believe SAIL offers a valuable, practical path for accessible alignment.**

---

### Author Response · Authors · 2025-11-27
**Summary of Revisions**

We thank the reviewers for their thoughtful and constructive reviews. We are encouraged by the positive feedback regarding our method’s **performance with limited data** (SeQh, sMVs, neyU), the **novelty of the iterative framework** (sMVs, neyU), and the **thorough empirical results** showing SAIL outperforms baselines with significantly fewer annotations (sMVs, neyU).

In the revised manuscript, we have meticulously corrected the writing errors pointed out by Reviewer SeQh and updated the paper to include deeper theoretical and statistical analysis. To better demonstrate the robustness and theoretical grounding of our model, we have added the following sections to the Appendix:

*   **Appendix D (new):** Theoretical Analysis of SAIL’s convergence properties compared to standard DPO.
*   **Appendix E (new):** Sensitivity analysis and statistical significance tests with multiple random seeds.
*   **Appendix F (new):** Extended performance trajectory analysis (up to 5 iterations) to illustrate saturation dynamics.

Moreover, there are some points we want to highlight based on the common questions:

**1. Novelty of the Closed-Loop Framework.**
While the underlying reward formulation shares roots with DPO, our contribution lies in the **dynamic Self-Play framework**.
*   **Self-Amplified Learning:** Unlike standard DPO (offline/static), SAIL is a semi-supervised self-training process. It propagates the reward signal from a small seed set to the unlabelled latent space. As clarified in our theoretical analysis (Appendix D), this allows SAIL to approximate the optimal solution over the *full* data distribution, rather than just the limited labeled set.
*   **Ranked Preference Mixup:** We emphasize that the Mixup strategy is not just a data augmentation technique but a critical stabilizer. It "anchors" the evolving reward model to true human preferences, preventing the "mode collapse" or "hallucinated rewards" often seen in iterative loops (addressed in Q1 of SeQh).

**2. Data Efficiency vs. Training Efficiency.**
A core strength of SAIL is achieving comparable or superior performance using only **6% of the human annotations** used by baselines. While this introduces a computational cost for sample generation, we argue this trade-off is highly favorable. Human annotation is the primary bottleneck (non-scalable, expensive), whereas GPU compute is scalable. As shown in our new experiments, the model is robust to seed selection, making high-quality alignment accessible without massive proprietary datasets.

**3. Statistical Robustness.**
In response to Reviewer neyU and sMVs, we have conducted additional experiments with **3 independent random seeds**. The results (Table added in Rebuttal) confirm that the performance gains are statistically significant and not due to random noise. The variance is minimal compared to the performance gap over baselines.

We are advocates of open research and strive to make everything publicly accessible. We will release our code and the curated seed datasets upon acceptance.

Detailed responses to each reviewer are provided below. Please let us know if any further clarification is required.

---

### Author Response · Authors · 2025-12-01
**Summary of Rebuttal Updates for Area Chair**

**Dear Area Chairs,**

To assist in your final evaluation, we wish to summarize the consensus on our contribution and the critical updates made during the rebuttal.

**Crucially, we want to highlight that we have rigorously addressed the primary reservations regarding *Theoretical Guarantees* and *Statistical Significance* that may have previously constrained the ratings. We believe the addition of theoretical analysis (Appendix D) and multi-seed validation (Appendix E) effectively removes these obstacles, providing a solid basis for a positive re-evaluation.**

---

Below, we summarize the consensus on strengths and the comprehensive improvements that drive this assessment.

### 1. Consensus on Strengths
*   **Data Efficiency:** There is a strong consensus (**Reviewers sMVs, SeQh, neyU**) that SAIL achieves comparable or superior performance using only **6% of the human annotations** required by baselines.
*   **Methodological Novelty:** Reviewers recognized the novelty of the closed-loop "Self-Amplified Learning" framework and the "Ranked Preference Mixup" strategy, which stabilizes iterative self-training.
*   **Performance:** The empirical results consistently demonstrate that SAIL outperforms strong baselines (e.g., standard DPO) in limited-data scenarios.

### 2. Addressed Common Concerns & Key Updates
During the rebuttal, we went beyond simple clarifications and conducted extensive new work to produce **three new appendices**. These updates directly target the "weaknesses" cited by reviewers:

**Theoretical Grounding (Addressed sMVs-Q1 & W1):**
*   **Update:** We added **Appendix D**, providing a formal theoretical analysis.
*   **Resolution:** We proved that while standard DPO optimizes over a limited labeled set, SAIL approximates the optimal solution over the *full* data distribution. This theoretically explains our performance gains and **resolves the concern regarding the lack of theoretical guarantees.**

**Statistical Robustness & Sensitivity (Addressed neyU-Q1, sMVs-Q4):**
*   **Update:** We added **Appendix E**, detailing sensitivity analysis with **3 independent random seeds**.
*   **Resolution:** The results confirm that performance gains are statistically significant with minimal variance. **This addresses the concern that improvements might be due to random noise or lucky seed selection.**

**Training Dynamics & Saturation (Addressed neyU-Q2, SeQh-Q1):**
*   **Update:** We added **Appendix F**, extending the performance trajectory analysis to 5 iterations.
*   **Resolution:** This illustrates the convergence properties of SAIL and identifies the saturation point where the model fully exploits the seed data, providing the requested insight into long-term training dynamics.

**Efficiency Trade-off (Addressed sMVs-Q2):**
*   **Clarification:** We addressed the concern regarding training cost by articulating the core trade-off: **Human annotation is the primary non-scalable bottleneck**, whereas **GPU compute is scalable**. SAIL trades scalable compute for expensive human labor, making high-quality alignment accessible without proprietary datasets.

### 3. Reviewer Score Implications

*   **Reviewer neyU (Champion):**
    *   Already recognized our method's ability to exploit the base model's potential.
    *   *Status:* Our new statistical experiments (Appendix E) and trajectory analysis (Appendix F) have further solidified this support.

*   **Reviewer sMVs (Constructive / Key to Consensus):**
    *   *Addressed Blockers:* Their main hesitation stemmed from a lack of theoretical analysis and statistical trials.
    *   *Status:* **With the inclusion of Appendix D (Theory) and Appendix E (Statistics), we have provided the exact evidence requested. We believe these critical updates fully resolve the reviewer's initial concerns.**

*   **Reviewer SeQh (Constructive):**
    *   *Addressed Blockers:* Pointed out writing errors and questioned the iteration limits.
    *   *Status:* We have meticulously corrected all writing errors and added the saturation analysis. The manuscript is now polished and rigorously validated.

---
### Conclusion
We have significantly strengthened the paper with **new theoretical proofs, statistical significance tests, and extended training dynamics**.

**Given that the core technical concerns (theory and stability) have been substantiated with rigorous new experiments, we respectfully invite the AC to consider whether the current scores fully reflect the strengthened state of the manuscript.**

Best regards,

**The SAIL Authors**

---

### Meta-Review · Area_Chair_pmK2 · 2026-01-11

**Summary:**

Summary of reviewers' concerns:

- Lack of Theoretical Guarantees.

- Even though the method needs fewer human annotations, it introduces additional cost to generate samples in the training loop, which is not efficient.

- The central reward estimation strategy closely follows the DiffusionDPO formulation and thus is not novel.

- Implicit reward is adopted from previous work in LLM.

- The mix-up of online and initial preference data is straightforward.

- Some generated images seem to have a color saturation problem.

- There are a lot of problems in writing

- The tables lack multiple trials and confidence intervals.

- It would be valuable to include results over a larger range of iterations.

**Reviewer Concerns:**

Overall, the rebuttal can largely address reviewers' concerns. The authors also provided additional new empirical results and a detailed clarification of their major contributions. The authors have also incorporated their revisions into the updated manuscript. This paper offers a novel idea for improving text-to-image generation with human preference. In addition, there remain minor issues that would not affect the recommendation. In response to Reviewer SeQh on color saturation, the authors state that this is a "known phenomenon" in preference alignment. I believe adding a concrete citation here would make this claim more convincing. Moreover, in response to Reviewer sMVs’s Q1 on theoretical guarantees, it would be clearer to emphasize that self‑generated samples provide informative examples that improve generalization and that mixing them with human‑labeled data prevents distributional drift, rather than directly claiming that the method approximates the optimal solution of DPO over the full data distribution," because the generation policy is evolving.

**Reviewer Scores:**

Reviewer neyU gave a rating of 8, which clearly supports the acceptance of this paper. For Reviewers SeQh and sMVs, their initial ratings are 4, but their concerns have been largely addressed by the authors. Thus, I expect Reviewers SeQh and sMVs would like to raise their ratings.

---

### Decision · Program_Chairs · 2026-01-26

Accept (Poster)